## Research Article

digital technology; community health worker; depression; psychosocial intervention; task-sharing

**Corresponding author:**
John A. Naslund;
Email: john_naslund@hms.harvard.edu

# Development of a digital program for training non-specialist providers to deliver a psychosocial intervention for depression: a formative study to support scaling up task-shared depression care in the United States

John A. Naslund[1] [iD], Natali Carmio[1], Sarah Taha[1], Margaux Amara[2], Sheena Wood[3], Anushka Patel[1] [iD], Sara Romero[1], Kyle Floyd[4], Brittany Meredith[4], Berta Rodriguez[5], Kelly Grajeda[5], Rebecca Brune[6], Andy Keller[5], Vikram Patel[1,7] [iD] and Katherine Sanchez[4]

[1]Department of Global Health and Social Medicine, Harvard Medical School, Boston, MA, USA; [2]David Geffen School of Medicine, University of California Los Angeles, Los Angeles, CA, USA; [3]South End Community Health Center, Boston, MA, USA; [4]Baylor Scott & White Health System, Dallas, TX, USA; [5]Meadows Mental Health Policy Institute, Dallas, TX, USA; [6]Congregational Collective, San Antonio, TX, USA and [7]Department of Global Health and Population, Harvard T.H. Chan School of Public Health, Boston, MA, USA

## Abstract

Task-sharing holds promise for bridging gaps in access to mental healthcare; yet there remain significant challenges to scaling up task-sharing models. This formative study aimed to develop a digital platform for training non-specialist providers without prior experience in mental healthcare to deliver a brief psychosocial intervention for depression in community settings in Texas. A 5-step development approach was employed, consisting of: blueprinting, scripting, video production and digital content creation, uploading digital content to a Learning Management System and user testing. This resulted in the development of two courses, one called *Foundational Skills* covering the skills to become an effective counselor, and the second called *Behavioral Activation* covering the skills for addressing adult depression. Twenty-one participants with a range of health-related backgrounds, including 11 with prior training in mental healthcare, completed the training and joined focus group discussions offering qualitative feedback and recommendations for improving the program's usability. Participant feedback centered around the need to make the content more interactive, to include additional engaging features, and to improve the layout and usability of the platform. The next steps will involve evaluating the training program on developing the skills of non-specialist providers and supporting its uptake and implementation.

## Impact statement

Across underserved communities in the United States and globally, there are significant gaps in access to quality mental health services. Task-sharing is a promising strategy for bridging these gaps; yet, innovative approaches are required to scale up the use of task-sharing. A key objective of this formative study was to develop a digital platform for training non-specialist providers without prior experience in mental healthcare to deliver a brief psychosocial intervention for depression. Importantly, following a systematic design process, this project resulted in the development of two courses: one called *Foundational Skills* covering the skills to become an effective counselor and the second called *Behavioral Activation* covering the skills for addressing adult depression. Additionally, participant feedback collected throughout this project helps to ensure that the training content will be relevant and acceptable for further pilot testing and implementation within the target community settings in Texas. This project represents an initial critical step towards adapting and implementing task-sharing models to increase access to proven psychosocial interventions for mental health problems in underserved settings in the United States.

## Introduction

Depression is a leading cause of disability in the United States, affecting roughly 8% of adults in any given year (Olfson et al., 2016). The resulting impact on individuals, their families and communities is substantial in the form of lost economic opportunity, caregiver distress and

increased risk of unemployment, homelessness, poverty, suicide and premature mortality (Walker et al., 2015). Alarmingly, less than one-third of adults with depression in the United States receive any treatment (Olfson et al., 2016). For patients who do receive treatment for depression, the vast majority – about 87% – are prescribed antidepressant medications, while only about 23% receive psychosocial interventions such as psychotherapy (Olfson et al., 2016). This is a major concern because in addition to strong evidence supporting the clinical effectiveness of psychosocial interventions (Cuijpers et al., 2007; Cuijpers et al., 2009; Cuijpers et al., 2021), patients overwhelmingly express a preference for psychosocial interventions over pharmacological treatment (McHugh et al., 2013). However, a critical challenge facing health systems is determining how best to scale up access to preferred psychosocial interventions given the formidable barrier of the shortage of specialist mental health providers (Dinwiddie et al., 2013; Lê Cook et al., 2013; HRSA, 2016; Larson et al., 2016; Andrilla et al., 2018). Innovative approaches are needed to expand access to proven psychosocial interventions, such as behavioral activation (Dimidjian et al., 2011), to address the significant gaps in access to depression care while recognizing the preferences of patients.

An important innovation in depression care supported by scientific evidence attesting to its effectiveness involves the delivery of proven psychosocial interventions by frontline providers without specialized training in mental health care delivery (Singla et al., 2017; Cuijpers et al., 2018; Raviola et al., 2019). This approach, commonly referred to as 'task-sharing' is supported by over 100 randomized controlled trials conducted across diverse settings globally, including low-income and middle-income countries (LMICs) where there are significant shortages in mental health providers (Barbui et al., 2020). Research also supports the effectiveness of task-sharing models for delivering mental health care in higher-income countries such as the United States (Hoeft et al., 2018; Singla et al., 2021), where many individuals similarly lack access to effective mental health services. In the United States, challenges with access are further exacerbated as mental health providers are inequitably distributed, with significant workforce shortages in rural areas (Larson et al., 2016; Andrilla et al., 2018) and gaps in access that are particularly pronounced for underserved racial and ethnic minority groups (Dinwiddie et al., 2013; Lê Cook et al., 2013).

Despite robust evidence supporting task-sharing models, there remain several challenges to scaling up and sustaining these approaches. One such challenge is the lack of scalable methods to train and ensure the clinical skills and competencies of non-specialist providers so that they can effectively deliver evidence-based psychosocial interventions. Digital technologies have emerged as promising tools for training non-specialist providers, as reflected in recent studies from Pakistan and India (Rahman et al., 2019; Muke et al., 2020; Nirisha et al., 2023). Digital training programs can reduce costs associated with travel, classroom space, and time and availability of expert trainers while offering participants the freedom to learn and acquire new skills at their own pace (Sissine et al., 2014). Online training programs are widely used in the United States for health worker training, typically in the context of continuing education and programs focused on basic skills for responding to the needs of patients with mental disorders (Sinclair et al., 2016; Jackson et al., 2018; Dunleavy et al., 2019). Training for non-specialist providers, such as certification for community health workers or programs focused on chronic disease management, are often available online or in a hybrid format consisting of a combination of remote and in-person training (National Association of

Community Health Workers, 2024; National Community Health Worker Training Center, 2024; Yeary et al., 2021; Zheng et al., 2021). It will be important to consider how to expand on the current widespread use and acceptability of remote training in the United States for specifically supporting non-specialist providers with gaining the skills to deliver evidence-based psychosocial interventions for depression.

This study aimed to design and develop a digital program for training non-specialist providers in the United States to deliver an evidence-based psychosocial intervention for depression. Specifically, we replicated prior efforts to digitize training content for use in rural India (Muke et al., 2019; Khan et al., 2020; Muke et al., 2020; Naslund et al., 2021), to adapt a program for training non-specialist providers in the delivery of a behavioral activation intervention for depression in underserved settings in Texas, United States. There are significant gaps in access to quality mental health care in Texas, which is made more complex to solve due to workforce challenges, given that over 80% of the state's 254 counties were designated as 'Mental Health Professional Shortage Areas' (Meadows Mental Health Policy Institute, 2016). Furthermore, our study adds to a growing number of efforts aimed at adapting task-sharing models for reaching underserved patient populations in the United States (Belz et al., 2024; Kanzler et al., 2024; Mensa-Kwao et al., 2024). We describe the stepwise process for the creation and digitization of the digital training program, followed by initial usability testing with a combination of non-specialist providers and experienced mental health providers, to collect feedback on the content to inform further refinements to the digital training curriculum.

## Methods

### Ethics

Institutional Review Boards at Harvard Medical School, Boston, Massachusetts and the Baylor Scott & White Health System, Dallas, Texas approved all study procedures.

### Evidence-based psychosocial intervention for depression

We adapted content from the Healthy Activity Program (HAP), an evidence-based intervention for depression developed and evaluated in India (Chowdhary et al., 2016; Patel et al., 2017). We selected HAP given its proven effectiveness in community settings (Patel et al., 2017; Weobong et al., 2017), and because it has been adapted for delivery by non-specialist providers across settings in India (Shidhaye et al., 2017; Shidhaye et al., 2019), Nepal (Walker et al., 2018; Jordans et al., 2019), Uganda (Rutakumwa et al., 2021) and Eswatini (Putnis et al., 2023). Furthermore, HAP employs behavioral activation, a proven and cost-effective intervention that has been widely used for treating depression (Cuijpers et al., 2007; Stein et al., 2021).

The intervention manuals for HAP are available open access from Sangath (see https://www.sangath.in/), a non-governmental organization in India, and have been previously digitized for training community health workers in rural India (Brahmbhatt et al., 2024; Khan et al., 2020; Muke et al., 2020). Given the success in adapting HAP for delivery by non-specialist providers across diverse settings, there is high potential to adapt this program for a US-based context. HAP consists of two manuals, one covering core skills to be an effective counselor (called *Foundational Skills*) and the second covering specific skills to deliver the intervention for

adult depression (called *Behavioral Activation*). The intervention is recommended as a first-line treatment for depression in the World Health Organization's (WHO) Mental Health Gap Action Program (mhGAP), which provides a roadmap for scaling up access to mental health care in low-resource settings (WHO, 2010). The intervention is delivered over 6–8 sessions and consists of content focused on psychoeducation, activity monitoring, activity structuring and scheduling, problem-solving, activation of social networks and behavioral assessments (Chowdhary et al., 2016).

### Stepwise development of the digital training platform

The development of the digital training platform modeled an approach previously employed in designing digital programs for community health workers in rural India (Khan et al., 2020; Shrivastava et al., 2023; Tyagi et al., 2023). This stepwise process draws from the educational literature, informed by the ADDIE (Analyze, Design, Develop, Implement and Evaluate) framework for instructional design (Obizoba, 2015), and by principles in human-centered design (Black et al., 2023). The ADDIE framework offers a systematic approach for designing and evaluating a training curriculum and instructional content, and has been employed to support the development of online health worker training programs (Patel et al., 2018). Specifically, we closely followed the systematic development approach previously employed in India and described in detail by Khan et al. (2020), whereby feedback can ensure the balance of fidelity and usability (Khan et al., 2020). For instance, feedback collected from experts is necessary to ensure alignment with the evidence-based intervention manuals while feedback collected from the target audience of frontline non-specialist providers can help to ensure that the content is relevant and acceptable (Khan et al., 2020). We developed the digital training content sequentially, beginning with *Foundational Skills* and then *Behavioral Activation* to ensure user feedback captured during the development of the first course could guide the development of the second course. Figure 1 illustrates this process, which involved gathering feedback and insights from content experts, non-specialist providers and health workers with experience delivering mental health care. These steps are detailed in the sections that follow.

### Step 1: Blueprinting

Consistent with prior development of digital training programs (Khan et al., 2020), the initial step involved creating a course 'blueprint' outlining the key learning objectives and core competencies covered in the training content. The blueprint serves as a course syllabus, outlining the overarching modules for the course and covering broad topics with a breakdown of the specific lessons and learning objectives covered within each module. The blueprint was developed through careful review of the training content while ensuring that key learning objectives for each module (taken from the HAP manuals) aligned with the skills and competencies that learners must ideally master following completion of the training program (Coderre et al., 2009). The blueprint was developed by two members of our team with clinical experience in mental health care and delivery of psychosocial interventions for depression, including clinical psychology and nursing. Next, the blueprint was finalized through review by three licensed clinical psychologists with extensive experience in the delivery of psychological interventions and training frontline providers. One of these expert reviewers was also involved in the early evaluation of behavioral activation for treating depression and in the development of HAP and therefore could offer an in-depth understanding of the content. Table 1 summarizes the final blueprint for the *Foundational Skills* and *Behavioral Activation* courses.

### Step 2: Scripting

Using the course blueprint as a guide, content from the manuals was written and adapted into scripts for the development of a series of short videos covering the lesson content. This process also involved cultural adaptation of the intervention content. For instance, the HAP manuals had previously been developed for use in India; therefore, cultural adaptations consisted of making the content relevant for a US context, such as changing terms used in role-play scenarios, such as 'market' to 'shopping mall,' 'tea stall' to 'coffee shop,' or 'temple' to 'church' or another community location. The use of short videos was informed by prior studies developing digital training programs for non-specialist providers, where videos spanning 3–8 min were most engaging without causing fatigue (Khan et al., 2020; Tyagi et al., 2023). The videos

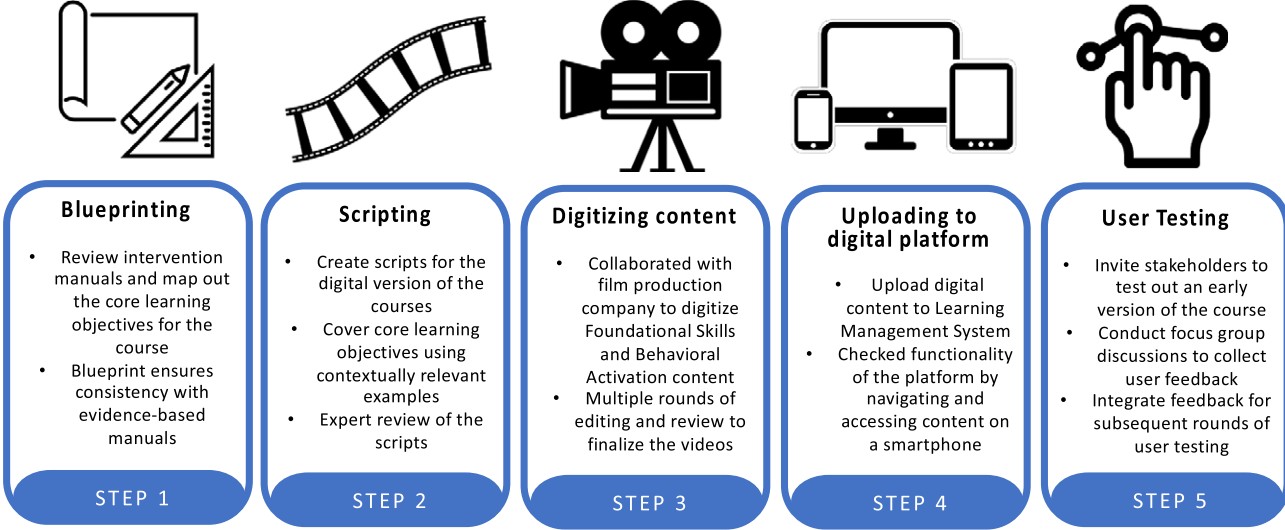

**Figure 1.** Overview of the stepwise approach to the development of the digital training program for frontline providers in Texas.

**Table 1.** Digital curriculum 'blueprint' for foundational skills and behavioral activation courses

| Foundational Skills (FS) | | |
|---|---|---|
| **Module** | **Topic** | **Learning Objectives**<br>*In this module, you will learn:* |
| | Roadmap to FS course | Module 1 - Module 8: Course titles and learning objectives |
| 1 | What is mental health? | • What constitutes mental wellness and illness?<br>• What are common mental disorders and their basic symptoms/signs?<br>• What are evidence-based therapies and how can they be applied to treat these disorders?<br>• Does counseling work or are medications the best treatment? |
| 2 | Assessment | • How to assess disorders using screeners with cutoff scores and clinical impressions?<br>• How to match a disorder to its evidence-based psychotherapy treatment? |
| 3 | What is counseling | • What is meant by counseling?<br>• What is the difference between counseling and a friendly chat? |
| 4 | Effective counseling relationships | • What is meant by an effective counseling relationship?<br>• What are the key skills for developing an effective counseling relationship?<br>• What are the different styles of counseling? |
| 5 | Creating a safe and effective counseling environment | • How to prepare ourselves for the counseling session?<br>• How to greet the client and introduce ourselves?<br>• How to talk about confidentiality?<br>• How to apply all these principles in different settings (e.g., by phone or by home visit)? |
| 6 | Managing high-risk situations | • How to manage suicide risk?<br>• How to conduct a suicide risk assessment?<br>• When to refer suicidal clients?<br>• How to apply effective skills and strategies for counseling a suicidal client?<br>• How to manage personal crises?<br>• What are the goals of crisis counseling?<br>• How to assess a client in a crisis?<br>• What are the steps in crisis counseling?<br>• How to help clients in bereavement?<br>• How to manage domestic violence?<br>• How to assess substance use disorder (levels of use) and determine whether it is a risk factor for suicide?<br>• How to provide basic motivational interviewing, use harm reduction strategies and know when to refer? |
| 7 | Social support | • Why do we need to involve social support in counseling?<br>• What are the situations in which we will need to involve social support in counseling?<br>• How to involve social support in counseling?<br>• What are the necessary precautions while involving social support in counseling? |

*(Continued)*

**Table 1.** (*Continued*)

| Foundational Skills (FS) | | |
|---|---|---|
| **Module** | **Topic** | **Learning Objectives**<br>*In this module, you will learn:* |
| 8 | Supervision, self-care, and wrapping up treatment | • How to keep in contact with clients?<br>• How to maintain ethical and professional standards?<br>• How to maintain boundaries with clients? |

| Behavioral Activation (BA) | | |
|---|---|---|
| **Module** | **Topic** | **Learning Objectives**<br>*In this module, you will learn:* |
| | Roadmap to BA course | Module 1 - Module 8: Course titles and learning objectives |
| 1 | Understanding depression | • What is depression?<br>• What are the symptoms of depression?<br>• How to screen for symptoms of depression?<br>• When and how to manage medication use for depression? |
| 2 | Introduction to BA | • What is BA?<br>• What are the phases of BA?<br>• How do you deliver each session of BA in a step-by-step manner? |
| 3 | Getting started with BA | • How to establish an engaging and effective relationship between a client and counselor?<br>• How to help clients understand the BA program?<br>• How to elicit commitment for counseling? |
| 4 | Learning together | • What are the goals of Phase II of BA?<br>• How to identify client values?<br>• How to identify activation targets? |
| 5 | Phase 2, getting active, and solving problems | • What is BA about and how to apply it to daily life?<br>• How to encourage activation?<br>• What are the barriers to activation and how to overcome these?<br>• How to help clients solve (or cope with) life problems? |
| 6 | Phase 3 and ending well | • How to help clients' review the BA model in general and specific actions that support the client's mood?<br>• How to help clients identify possible challenging future situations?<br>• How to help clients make a plan to deal with such situations using the skills they have learned? |
| 7 | Useful strategies for specific problems | • What are the ways to deal with common problems that can make it more difficult to get active and solve problems?<br>  ○ No motivation to do activities<br>  ○ Thinking too much<br>  ○ Feeling anxious or tense<br>  ○ Problems with people close to the client<br>  ○ Societal stressors (e.g., anxiety about the future)<br>  ○ Difficulties with sleep<br>  ○ Using tobacco, alcohol and/or marijuana |
| 8 | Course summary | • Review of the phases of BA<br>• What to do when seeing a client for the first time? |

followed two format types in each course: (1) didactic instructional videos and (2) role-plays between patients and counselors. In didactic videos, the training content is presented to learners by counselors with extensive experience utilizing these skills, who discuss the definitions and examples associated with clinical concepts. The scripts created for role-play videos illustrated the implementation of various clinical skills in practice by showing realistic counselor-patient interactions, with a focus on key aspects of the therapeutic encounter. Examples include how counselors provide psychoeducation, manage crises (e.g., suicidal ideation), motivate change and problem-solve barriers with patients to homework completion. The video scripts included dialog and screen directions, describing where the scene takes place, props used and other production-related instructions for use by the director and video production team. The scripting process involved creating a list of the different actors needed to play various characters in the videos, including the counselors, patients or family members/significant others, and the characteristics of these actors (such as age, sex, race/ethnicity and other notes) and when they appear in each lesson. The screen directions were used to reinforce learning concepts, such as keywords appearing on the screen when the counselor is demonstrating a particular skill in the role-play. All scripts were written and/or adapted by our team, before inviting expert feedback from the three licensed clinical psychologists who partnered on this project. All videos developed as part of this training involved the use of actors, as opposed to actual clinicians or patients. This was mainly to streamline and standardize the production process given the need to adhere to the scripted content and project timeline.

### Step 3: Video production and digital content creation

Our team reached out to multiple video production companies and selected two companies based on the criteria of quality and timeline. One company was hired to support video production for the *Foundational Skills* course and the second company was hired to support video production for the *Behavioral Activation* course. Our team worked closely with the video production companies to review and edit the scripts, collect necessary props (e.g., worksheets, brochures, etc.), and select actors to play the different characters (e.g., counselors, patients, family members/significant others, etc.). The videos generally range in duration from 3–8 min to facilitate access from a smartphone device and to retain learner engagement. Production time for each course required approximately 6 months, consisting of 2 months for pre-production, 1 month for filming, and 3 months for video/sound editing, and allowing for 3 rounds of review and feedback. Actors were used in all the videos to represent patients and counselors, as this allowed for expedited development and production.

To reinforce learning, the videos were supplemented with digital content, including brief text summaries of key points in each video. Knowledge checks via multiple-choice questions served to reinforce concepts and were interspersed throughout each lesson to help learners stay engaged. The correct responses to the knowledge check questions are available for learners at the end of each module. Supplemental materials were provided at the end of each module, including articles, YouTube videos, graphics or strategies curated from reputable sources (e.g., American Psychological Association, Centers for Disease Control), peer-reviewed academic research papers, intervention worksheets or text summaries created by our team. These supplemental materials were initially reviewed internally by members of our team with expertise in clinical psychology, before undergoing review by the 3 external experts who previously

had reviewed the project blueprint described above. Drafting, reviewing, and finalizing the digital content through expert review was completed concurrently with the video production phase.

### Step 4: Uploading digital content to the learning management system

The digital platform used to host the course content (i.e., the videos and other digital content) is referred to as a Learning Management System (LMS). The LMS allows learners to navigate the course content and offers features including *content management tools* that allow for the creation or upload of content and assigning it to specific individuals or groups, *assessments and testing* that allow learners to complete questionnaires and progress through the course, *mobile optimization* that ensures access to the content on a smartphone app, and *reporting* to enable tracking learners' progress completing the course. Before the digital content can be uploaded onto the LMS, it must first be formatted using an e-learning authoring tool. We used the Articulate authoring tool to combine the digital content (i.e., videos, audio, text and graphics) for export as SCORM (Sharable Content Object Reference Model) files for uploading onto the LMS. The authoring tool facilitates formatting written content, adding knowledge check questions and incorporating supplemental materials. An important advantage of using the Articulate authoring tool to generate SCORM files is that these files can be uploaded to most available LMS platforms, allowing us to seamlessly move the digital course to other platforms. This is an important consideration for scaling up such training programs, as the digital content could be uploaded to an organization's existing LMS platform which could support future uptake of the training. In this project, we used the Cornerstone On Demand (CSOD) Learning Management System mainly because it is a widely used platform in many health systems across the United States and includes a cloud-based server which can accommodate a large number of users accessing the training program content. The full lesson content was finalized in Articulate and uploaded to the LMS for final review and user testing.

### Step 5: User testing of the digital training platform

We invited health workers and trainees with varying levels of experience to test the first iterations of the digital content. The goal was to assess the usability of the digital training, to identify and address potential challenges with the platform and engaging with the content, and to solicit recommendations for additional features or materials that could be included. We employed a convenience sampling approach to recruit participants from contact lists of the project collaborators. Recruitment involved sending out informational emails with details about the project and inviting interested individuals to reach out to learn more and to participate in the training. We prioritized reaching health workers and other individuals with prior experience delivering mental health care. This was to ensure that we received feedback from a target group that could draw from their own prior training and clinical experiences to offer insights for guiding modifications and improvements to the content. Participants did not receive any payment for completing the training.

Participants completed written informed consent, followed by a brief questionnaire consisting of demographic details and prior work experience and training in mental health care and were then instructed to complete the digital course. We informed participants that our team was available at any time by email or by phone during

regular work hours in the event there were any concerns encountered during the training, such as questions about materials or technical challenges. Participants were then invited to join a focus group discussion over the Zoom teleconferencing platform to share their feedback. A member of our team facilitated the focus group discussions using a semi-structured interview guide (see Supplemental Material) to collect participants' impressions about the training, their experiences, and comments on the ease of navigation, interacting with the content, wording of the content, ability to access the platform on various digital devices (i.e., smartphones, tablet and laptop), structure and layout of the course and relevance and usefulness of the training for their current work. Participants were also requested to provide suggestions about ways to improve the course. Moreover, this formative research process provided the opportunity to address technical issues such as difficulties related to logging into the platform or connectivity issues.

### Qualitative data analysis

The focus group discussions were audio-recorded and transcribed by two research assistants. A third research assistant reviewed the transcripts and listened to the audio recordings to ensure accuracy and that no key points were omitted. Our analysis was guided by a framework analysis approach (Gale et al., 2013), which involves a content analysis technique often used in applied qualitative research and allows for the inclusion of pre-determined topic areas (Ward et al., 2013). We selected this methodology to leverage the qualitative data for better understanding the usability of the digital training platform while capturing insights from participants that could inform improvements to the training format and content. Two members of our team read the transcripts and independently coded participant feedback according to three broad categories reflecting the digital platform navigation, the course layout and the instructional content covered in the course. Our interview guide had questions covering these three broad topic areas. After reading

and coding the transcripts, these two team members met to review each other's coding and to reach a consensus on a summary of the key points reflected in participants' comments about the training program and content. Next, they presented the broad summary to the larger research team to discuss the different recommendations, make decisions about what recommendations could be addressed, and consider the best approach for improving the layout and functionality of the training program in preparation for further evaluation and implementation.

## Results

### Digital training program

The development process required approximately 6 months for each course (12 months in total). The final digital training content was uploaded to the CSOD LMS, accessible from a smartphone app or web browser, where a course administrator could manage the course, track participant progress and send notifications. The final training consisted of 8 modules for *Foundational Skills* and 8 modules for *Behavioral Activation*, as outlined in Table 1. Each module covers a broad topic and consists of a series of short lessons that align with the specific learning objectives for that overarching module. For *Foundational Skills*, there are 8 modules covering 37 lessons, each using a combination of video lectures and role-plays (53 videos in total) and includes a series of assessment questions covering the content presented in the videos. Similarly, for *Behavioral Activation*, there are 8 modules consisting of 34 lessons with video lectures and role-plays (55 videos in total) and supplemented with assessment questions to reinforce the key concepts and learning objectives. See sample content presented in Figure 2, as well as an illustration of the user interface on the CSOD platform in Figure 3.

The videos ranged in duration from 3 to 8 min. Various approaches were used to make the video-based content interesting

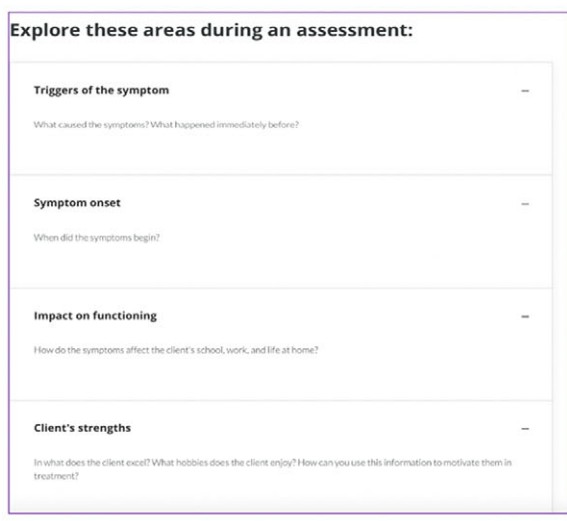

**Figure 2.** Sample screenshot showing content from one of the modules from the Foundational Skills course.

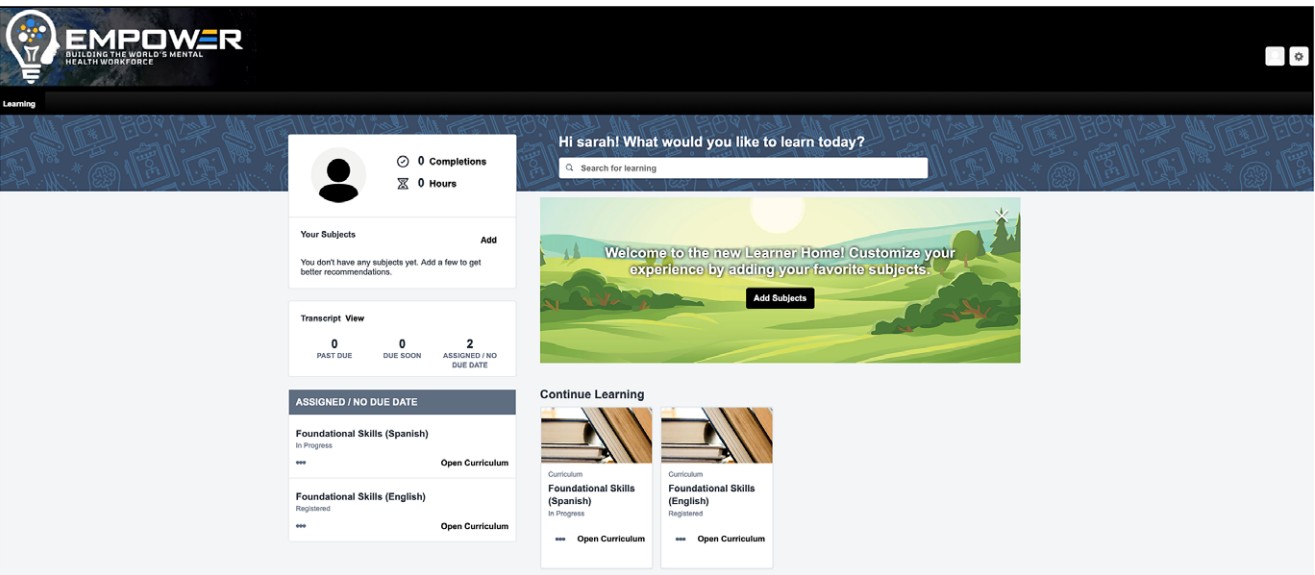

**Figure 3.** Learning management system user interface.

and engaging, including the use of illustrative cutaways of counselor-patient interactions in clinical scenarios, presentation of text on the screen to reinforce core concepts, assessment questions including multiple-choice questions, drag-and-drop response options, and flashcards to accompany the videos and to support knowledge comprehension. Role-play videos were a critical component of the training program to offer a demonstration of the use of specific counseling skills and interaction between patients and counselors during delivery of counseling sessions. The time required to complete both courses, along with assessment questions and activities embedded in the digital platform is approximately 20-40 h (about 10-20 h for each course). We determined the time to complete the course by having student interns who were not involved in designing the course, to ensure no prior exposure to the content and record the time necessary to fully complete all the course materials. The course materials are accessed sequentially, meaning the modules need to be completed in order and access to the *Behavioral Activation* course is enabled after completing the *Foundational Skills* course. However, learners have the option to revisit modules and retake any assessment questions or activities as often as they like.

### Usability testing

We contacted 75 individuals through email invitations to try out initial versions of the digital training. Of these individuals, 26 consented and agreed to take the course. In total, 21 participants (out of 26; 81%) completed *Foundational Skills*, before joining a 1-h focus group discussion. The demographic characteristics of these 21 participants are summarized in Table 2. Over half (N = 11; 52%) of participants were mental health providers, consisting largely of individuals with experience in clinical psychology. All 21 participants who completed *Foundational Skills* were invited to complete *Behavioral Activation*, of which 12 completed the course (out of 21; 57%), and 9 joined a second 1-h focus group discussion. There were 4 focus groups for *Foundational Skills* (N = 21 participants) and 2 focus groups for *Behavioral Activation* (N = 9 participants). There was about a 1-month delay between completing *Foundational Skills* and starting *Behavioral Activation* to allow our team time to complete the first round of focus group discussions and to integrate participant feedback accordingly.

### Qualitative findings

Participant feedback from focus group discussions was grouped into three overarching topic areas to inform modifications to the digital training, including: platform navigation, course layout and content. Participants offered specific suggestions and recommendations for areas for improvement related to the user-friendliness and course content. Participants were generally impressed with the courses and indicated that they found the content useful, engaging and professionally designed. Participant feedback on *Foundational Skills* is summarized in Table 3, and participant feedback on *Behavioral Activation* is summarized in Table 4. For course layout, among participants who completed both courses, they tended to prefer *Behavioral Activation* over *Foundational Skills* because of the inclusion of more interactive features, such as being able to scroll through the content, rather than having to click each time for new content to appear on the screen. For navigation, participants did not appear to encounter any challenges and expressed satisfaction with the ease with which they could progress through the material. For the content, participants enjoyed the real-life role-play scenarios between counselors and patients, reflected across both courses. Participants consistently mentioned that they had not previously encountered courses of similar design and comprehensiveness.

Participants offered suggestions for improving the courses, such as including a roadmap at the beginning that incorporates the goals of the course and a tracker showing the time remaining so that participants could anticipate the time required while completing the training. There were also recommendations to include additional modules with role-play scenarios when patients or counselors encounter a problem during a counseling session, more variation in the types of knowledge assessment questions after each lesson (i.e., adding matching or short answer response questions) and ensuring consistency in the difficulty level as some of the modules were considered easier to complete relative to others. Participants also highlighted that the opportunity to engage in

**Table 2.** Participant demographic characteristics

|  |  | N = 21 | |
| --- | --- | --- | --- |
| Characteristic |  | Mean | SD |
| Age, *Mean (SD)* |  | 47.1 | 11.2 |
|  |  | N | % |
| Age | 26–35 | 9 | 42.8% |
|  | 36–45 | 3 | 14.3% |
|  | 46–55 | 4 | 19.0% |
|  | 56–65 | 3 | 14.3% |
|  | 66–70 | 1 | 4.8% |
|  | Preferred not to respond | 1 | 4.8% |
| Gender | Female | 18 | 85.7% |
|  | Male | 3 | 14.3% |
| Race | White | 15 | 71.4% |
|  | Black or African American | 2 | 9.5% |
|  | Asian | 3 | 14.3% |
|  | Preferred not to respond | 1 | 4.8% |
| Ethnicity | Hispanic or Latino | 4 | 19.0% |
|  | Not Hispanic or Latino | 17 | 81.0% |
| Education | Some college or certificate program | 4 | 19.0% |
|  | Bachelor's degree | 6 | 28.6% |
|  | Master's degree | 9 | 42.8% |
|  | PhD or doctoral equivalent | 2 | 9.6% |
| Mental Health Experience | Mental health provider | 11 | 52.4% |
|  | Non-mental health provider | 10 | 47.6% |
| Profession | Clinical psychology | 8 | 38.1% |
|  | Nurse | 6 | 28.6% |
|  | Community health worker | 4 | 19.0% |
|  | Social worker | 1 | 4.8% |
|  | Administrator | 1 | 4.8% |
|  | Student | 1 | 4.8% |

active learning would be helpful, with the use of checklists that they could fill out while navigating the content and including a manual with a glossary of clinical terms. Participants also made stylistic suggestions such as adding subtitles for the videos, which were considered useful to read along with the course, titles for modules and supplementary materials after each module rather than at the end of the course. Participants commented on their encounters with various technical issues such as the videos abruptly ending or technical glitches that made it difficult to navigate and complete the training. Participants' recommendations and resulting modifications to the digital training are summarized in Table 5.

## Discussion

This study involved the step-wise development of a digital program for training non-specialist providers in the delivery of a brief behavioral activation intervention for depression in the United States. Use of digital programs for training non-specialist providers offers important advantages for achieving scale, by overcoming barriers to participating in more costly classroom-based instruction or in-person workshops (Khanna and Kendall, 2015), and allowing participants to complete the training at their own pace. While using technology for the training of health workers is not new, there have been few efforts aimed at leveraging *fully remote* digital programs to develop skills and competencies of non-specialist providers in the delivery of psychosocial interventions in community settings (Frank et al., 2020). If we conceptualize training programs on a spectrum of least to most scalable, fully remote digital training programs represent some of the most scalable because the work is 'front-loaded' to create a course that later requires minimal changes (e.g., context-specific guidelines can be tailored to facilitate implementation). Our findings add to a limited, yet growing, number of studies of sustainable approaches for building the mental health workforce in US settings, including studies involving a combination of web-based and peer-led training programs (German et al., 2018), technology-enhanced training in cognitive behavioral therapy (Kobak et al., 2017), and online training in a psychosocial intervention augmented with an online learning collaborative supported by expert clinicians (Stein et al., 2015). We collected participant feedback to inform modifications to the training platform and content, which can potentially improve overall usability and overcome challenges with existing online training programs, such as low user engagement, high attrition and limited acquisition of new knowledge and skills (Santos et al., 2014; Conte et al., 2021; Hill et al., 2021).

While we replicated prior digital training development work reported in India (Khan et al., 2020; Shrivastava et al., 2023; Tyagi et al., 2023), our study adds to a small but growing number of studies aimed at leveraging and adapting successful global mental health efforts from LMICs for advancing the use of task-sharing models in the US (Belz et al., 2024). Despite robust evidence in support of task-shared mental health interventions in LMICs (Barbui et al., 2020), there remain few reports describing the systematic process for adapting these programs for use within a US context, as well as a lack of rigorous evaluation studies focused on underserved communities in the US. Therefore, our study represents a novel contribution to the field, and this formative work opens the door to new avenues to scale up proven psychosocial interventions in communities facing rising mental health burden and significant workforce shortages.

Our study complements the mounting emphasis on considering frontline non-specialist providers such as community health workers as being essential for bridging gaps in access to mental health services (McBain et al., 2021). Importantly, non-specialist providers are oftentimes ideally positioned to address social disparities, meet the needs of vulnerable patient groups and even reach patients in settings outside the formal health system (Knowles et al., 2023). This is critical in the context of mental health services, where concerns pertaining to stigma, mistrust of the health system, and barriers due to socioeconomic status, culture or language create significant challenges for addressing the mental health needs of at-risk communities (Wong et al., 2017; Pescosolido et al., 2021). Non-specialist providers, such as community health workers, can play a key role in addressing mental health challenges in historically underserved communities, as reflected in a preliminary study consisting of qualitative interviews with community health workers in Southern Texas near the US-Mexico border region (Garcini et al., 2022). The community health workers emphasized barriers to

**Table 3.** Summary of participant feedback on the Foundational Skills course

| Category | Codes | Representative Quotes |
|---|---|---|
| Platform Navigation | Physical experience (e.g., scrolling through, accessing modules, etc.) | "…liked how easy it was to use the course, easy to stop and start. Also tried on the phone, was sometimes frustrating, had to scroll around to hit next, but was good to see what the experience was like to use on a phone, was more distracting to do on her computer" – *Participant 1* |
| | Accessibility | "It would be good if there was an outline or transcript to review… being able to click through a transcript allows us to know which video covered what" – *Participant 2* <br> "It would be helpful to have closed captioning as well, would allow an even wider audience to take the course" – *Participant 1* |
| Course Layout | Timeframe and scheduling | "The time was appropriate for the content… but smaller chunks to really understand the info to digest and come back otherwise the learner can be fatigued; but total time was good and not excessive" – *Participant 2* <br> "It would be good to have sections and to know how many minutes are left; it was clunky to get out and get back in; an easier way to improve accessibility" – *Participant 10* |
| Content | General content | "I love the examples during the modules; does not just explain the concept but it tells how to help the patient to come up with the plan…what conditions to think about to bring to the supervisor and how to communicate it" – *Participant 5* <br> "I appreciated not only the context of why using each approach; but also, the information and it was not didactic but also the context of the why and the how. The course gave you phrases and words to use" – *Participant 10* |
| | Specific subjects (e.g., domestic abuse, high-risk situations, navigating telehealth, etc.) | "It could be useful to talk about what a risk assessment looks like during a telehealth visit" – *Participant 8* <br> "Felt like there wasn't COVID mentioned, liked that there was telehealth but could be useful to talk specifically about COVID" – *Participant 8* <br> "Depth on PTSD did not realize there are more levels to that which usually only PTSD was covered so that was really interesting" – *Participant 12* |
| | Videos | "The videos were exceptional. There was diversity in the actors, everyone had a different approach, and there were a lot of examples of how to ask about one thing. I only figured out what I should pay attention to at the end of the video, and it might be helpful to know this at the beginning."– *Participant 4* <br> "Subtitles should be added." – *Participant 2* |
| | Assessments | "It would be helpful to have the explanation after the questions" – *Participant 8* <br> "I wanted to apply knowledge more than recognize multiple choice. Make your own checklist or write your own thing, or questions with more typing and allow to personalize, and get application of it somewhere else. The role-plays are engaging and encouraging so want to try that" – *Participant 4* |
| | Supplemental materials | "Could be helpful to have downloads of the scripts"– *Participant 8* <br> "Integrate links from supplemental materials into the module or have a blurb saying go to this material" – *Participant 15* |
| | Relevance to work | "Would be good to include what you gain from [the course], what your goal is at the end, what it means to you to be a counselor… this happened later [in the course] and it would be great to bring it forward and have that engagement sooner" – *Participant 4* <br> "Enjoyed the role-plays, made it feel more real, seeing the interactions between the counselors and the patients. The role-plays felt like a realistic approach, and not reading a script" – *Participant 6* |
| | Accessibility of the training content for non-specialist providers and newcomers to mental health delivery | "If the target audience for the video is community health workers, have a video talking directly to the audience about fears/reservations of delivering counseling. As a clinical therapist, it can be very challenging, I can imagine how challenging it can be for someone who didn't go to school for it?" – *Participant 18* <br> "Are we assuming some basic understanding of mental health knowledge before coming into the course? Some language might need to be softened, for example what is evidence-based care? Also, could be useful to talk about what a risk assessment looks like during a telehealth visit" – *Participant 8* <br> "If going to start this work then it is great. A first glimpse of what can be done and what to explore, it is not overly clinical and technical. There is more to come, and it is a lot of info if brand new, and hard to absorb it all if you have no experience in it" – *Participant 3* |

addressing mental health concerns in their communities, including limited training opportunities and emphasized the need for needs-based training programs to build their skills in mental health care (Garcini et al., 2022). Therefore, the training program developed in this study could bridge this gap and support capacity building of community health workers working in underserved settings.

## Limitations

Several limitations of this study warrant consideration. While we achieved the goal of recruiting both specialist and non-specialist providers, the small sample size and reliance on a convenience

sampling approach make it difficult to generalize our findings. While the training was designed for use with non-specialist providers, over half of the participants in this study identified as mental health providers. As such, it will be critical to ensure further engagement of non-specialist providers, such as community health workers, to better understand the acceptability of the training and potential for integration and uptake as part of their current work responsibilities. The focus on evaluating the usability of the digital training represents another limitation, as it is not possible to confirm whether the digital training is effective in developing participants' skills and competencies to deliver the depression intervention in practice. There was also high attrition, where of

**Table 4.** Summary of participant feedback on the Behavioral Activation course

| Category | Codes | Representative Quotes |
|---|---|---|
| Platform Navigation | General navigation | "I thought it was easy to follow and I didn't have any trouble." – *Participant 2*<br>"Once the module ended, you had to manually click out and go back to the main page where it showed progress, then had to activate the next module for the next lessons to show up. It wasn't difficult but it would be nice if there was more automation" – *Participant 9*<br>"In my ideal world, I could scroll through the content *while* watching the videos. That's when the supplemental material makes sense to me. I think there needs to be prompts in the video like 'look below to follow along'" – *Participant 1*<br>"It would be great if we could improve the navigation from the curriculum page to each individual module" – *Participant 15*<br>"I appreciated that the videos introduced the content present in the rest of the lesson, but I almost missed the stuff you have to scroll through at the bottom after the videos" – *Participant 9* |
| Course Layout | General layout | "In Foundational Skills, I had many minor comments but I thought that Behavioral Activation was more polished, especially in the review process. Having live enactments was very nice, and it was nice to see real people presenting the content." – *Participant 2*<br>"The esthetics [in Behavioral Activation] are better than Foundational Skills; after I finished the training, I had to take time to figure out how to go back to the modules and figure out the videos but overall this training is better." – *Participant 15*<br>"It would be helpful to have more of a time outline, you want learners to retain the information and not just keep clicking; make things accessible and interactive for people actually in the field" – *Participant 8* |
|  | Transitions | "There's a lot of content in each lesson, but I liked starting off with the videos rather than written content." – *Participant 9*<br>"Within resources, you had to click on the links and then they would pop up (YouTube videos); it was all packed in there. I appreciated that, but it wasn't exactly intuitive and needs additional notation." – *Participant 9* |
| Content | Course content | "I felt like everything was very clear even though I am not trained in mental health at all" – *Participant 20*<br>"I think it gives a great overview, but I wouldn't say that people who take this course would be ready to counsel the next day." – *Participant 18*<br>"I thought the videos were well done and the format of explaining the technique and then showing them going in with the patient was really good." – *Participant 15*<br>"The content itself was very interesting, but if a user doesn't have an inherent interest and isn't self-paced/ independent, will they leave with all the skills they need or will they skip through and that is dangerous to implement" – *Participant 8* |
|  | Supplemental material | "I viewed supplemental material as optional, I usually wouldn't go through it unless it seemed very interesting or I needed additional resources. I could see how learners may not know they need the supplemental material until they are actually in practice and wouldn't necessarily seek it on their own." – *Participant 2*<br>"The supplemental materials were in image format and not PDFs. It would be helpful to have pdf versions so learners can print it and have a document with a compilation of all the supplemental materials." – *Participant 18*<br>"It would be helpful to know it is there in case we need it later on and to have it separately and organized by lesson" – *Participant 9* |
|  | Relevance to work | "I appreciated that the training mentioned COVID-related support; it is a big issue with patients now and further training with that and possible group therapy would be beneficial. I met with people who are really hurting. They lost so much and this loss is chronic- for the rest of their lives. I appreciated that the training included something so obvious to what we are dealing with right now" – *Participant 20*<br>"People who do not have formal training rely on the manual and the videos. It would be ideal to have a manual to learn the technique and another where they can open the manual and go through the module that day and be able to accomplish what needs to accomplish that session" – *Participant 18*<br>"It was a good refresher; it's not a modality I use often, but fidelity to that modality. It was very helpful to bring back core skills and to put myself in trainee's shoes, now learning a specific advanced skill" – *Participant 9*<br>"I liked how the training clarified ways to teach your clients skills; I've used it in at work the past week and it was very helpful" – *Participant 19* |

the 21 participants who completed the *Foundational Skills* course, only 12 went on to complete the *Behavioral Activation* course, raising concerns about the feasibility of the training and potential for uptake. The collection of additional usability metrics could help to better understand the reasons for discontinuing the training and inform opportunities to promote engagement or offer tailored support to learners. Ultimately, we acknowledge that the high attrition could be the result of the significant time commitment required to complete the training. We also observed dropout between courses, which could be further attributable to the roughly 1-month lag between completing *Foundational Skills* and starting *Behavioral Activation*, thereby suggesting that participants who went on to complete both courses were likely highly motivated

and interested in the content, and therefore, may have expressed more positive feedback about the training. This highlights an important area for our team to consider as we seek to roll out the training, as the time required will likely emerge as a barrier to implementation. Frontline non-specialist providers generally have limited time availability and already experience heavy workloads and without additional compensation or being provided adequate time in their workplace, there will be challenges to integrating this training in health systems in the US and elsewhere.

We emphasized the importance of capturing user feedback about the usability of the training, making sure to recruit a combination of non-specialist and specialist providers. However, we recognize that non-specialist providers were not engaged in the

**Table 5.** Summary of participant recommendations and resulting modifications aimed at improving the digital training program

| Recommendation | Response |
| --- | --- |
| Present an overview of all the content at the beginning before learners take the course | Added a roadmap that includes the learning objectives of all the modules to the beginning of the course |
| Incorporate a different format for the knowledge checks at the end of each lesson | Reformatted the knowledge checks to include a variety of questions aside from only multiple-choice questions such as matching, true/false and free response questions. Correct responses to the knowledge checks are available to participants after completing each module. |
| Provide access to supplemental materials by module rather than at the end of all modules | Included access to supplemental materials pertaining to each specific module right after all the lessons for the following module were completed |
| Ensure active learning instead of having learners go through slides | Adding flashcards to ensure that participants are internalizing the materials |
| Include additional information to reflect the COVID–19 circumstances and impact on mental health | Addressed the long-term consequences of the pandemic on mental health as part of the Behavioral Activation course |

early stages of scripting the content and reviewing the blueprint. This is partly because the content was adapted from the digital training developed and implemented in India, which had already undergone an extensive process of engaging community health workers in the development process, as well as time and funding constraints. To mitigate this limitation, we will pilot-test the training with a group of target non-specialist providers, and make additional modifications to the content as needed. One advantage of the digital platform is the flexibility to make modifications to the content and program layout. It is also important to note that while the training was adapted for use in a US context, our goal was to ensure that the training could be generalizable for use in as many settings as possible, meaning that the content is not tailored to specific cultural or demographic groups or specific geographic areas. For instance, an important future direction will be to evaluate the relevance of this training for use within rural settings and capture the perspectives of a wide range of non-specialist providers, which could cover nearly anyone without formal training in mental health care, about the relevance of this current program in their work. For example, in addition to community health workers, suitable participants for completing this training program could include members of congregations of churches, veteran serving organizations, college students, and ordinary people in community settings who may be ideally positioned to respond to the mental health needs of others.

## Future directions

It is important to note that this course represents only an initial step as part of training non-specialist providers and that it would be unlikely that after completing such a course someone would be prepared to deliver care without first having a chance to practice these skills under the supervision of an experienced clinician. This

is a critical observation, and it is essential to recognize that this is the first step in a broader journey towards achieving competence and ensuring progression from learning the content, to practicing the skills in real-world settings, and ultimately, mastering content and delivering high-quality care and improving patient outcomes (Patel et al., 2022). Access to ongoing supervision and opportunities to practice the newly learned content and skills must be considered a necessary extension to the current digital training program. For instance, newly trained non-specialist providers will need to apply what they have learned with support from experienced mental health clinicians, followed by independently seeing patients while engaging in routine supervision. Supervision of non-specialist providers in the delivery of psychosocial interventions, which can be completed remotely or in-person is essential to ensure quality, identify challenging cases for referral and offer additional opportunities for learning, further developing skills and knowledge and avoid the risk of burnout and exhaustion (Singla et al., 2020; Singla et al., 2024). Future studies are needed to examine such questions by linking learner milestones, engagement and knowledge scores to clinical competencies evidenced in performance metrics (e.g., role-play-based supervision and real-patient interactions) and care metrics (e.g., patient outcomes) (Singla et al., 2023).

There are additional critical steps to build on this formative research, beginning with the need to ensure that the training program is effective and can support the development of skills and achieving competency, which could be assessed using a standardized approach such as EQUIP (Kohrt et al., 2025). With widespread consensus that non-specialist providers, such as community health workers, are ideally positioned to reach underserved communities in the US experiencing significant mental health disparities (Barnett et al., 2018a), including racial and ethnic minority groups (Barnett et al., 2018b), there is a need to further explore the relevance of this training in these communities. For instance, given that this study was focused in Texas, a key next step will be considering the relevance of the training for reaching Hispanic communities near the US-Mexico border, and determining what additional cultural, linguistic and contextual adaptations to the training content are needed. Furthermore, rigorous evaluation of the costs required to deploy the training and to successfully support non-specialist providers with acquiring the skills and knowledge to deliver behavioral activation represents an essential future research direction. Understanding the cost implications will be critical for informing health systems and other key stakeholders about the resources required to build workforce capacity to address depression and to sustain these efforts in practice.

**Open peer review.** To view the open peer review materials for this article, please visit http://doi.org/10.1017/gmh.2025.5.

**Data availability statement.** All de-identified data in this study is available from the authors upon request.

**Author contribution.** JAN wrote the first draft, oversaw the project activities and made revisions to the manuscript based on feedback from co-authors. NC, ST, MA, SW, AP, and SR supported the project activities, data collection, analysis and interpretation and contributed to the writing and revision of the manuscript. KF, BM and KS led participant recruitment at the study sites, supported data collection and contributed to revising the manuscript. BR, KG and RB contributed to revising the manuscript. AK and VP supported the acquisition of funding for this work, provided oversight of the project activities, and contributed to revising the manuscript. KS provided oversight for the project activities and contributed to revising the manuscript. All authors approved the contents of the final manuscript and decided to submit for publication.

**Financial support.** This study was supported by grant funding from the Surgo Foundation, a gift from NM Impact Ltd., and the Lone Star Prize awarded by Lyda Hill Philanthropies. This project was also supported by a grant from the National Institute of Mental Health (U19MH113211–01). The funders played no role in the conduct of the study, analysis and interpretation of the findings and decision to publish.

**Competing interest.** The authors report no conflicts of interest.

**Ethics statement.** The institutional review boards at Harvard Medical School, Boston, Massachusetts and the Baylor Scott & White Health System, Dallas, Texas, approved all study procedures and human subjects activities in this project.

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
