## [Reviewer Report]

This paper describes the development of a digital training platform for non-specialist providers providing a psychosocial intervention. As non-specialist delivered psychological interventions are increasingly scaled, training programs that can also scale will be critical, yet training models for non-specialists remains an understudied area. Further, research on non-specialist delivery of psychosocial interventions is less developed in the United States than in low resource global settings to date, yet non-specialists are increasingly being deployed for psychological interventions in the US. I very much appreciate the focus of this paper, am looking forward to seeing any future outcomes papers, and have made some recommendations below to further refine it below.

- Introduction

o Where you discuss replicating prior efforts, it would be appropriate to reference some of the literature on reverse innovation/bidirectional learning in global mental health.

- Methods

o Can you clarify how the adaptation process described on page 5 and the blueprinting and scripting process described on page 6-7 relate to one another? Was cultural adaptation conducted before or during blueprinting and scripting? What were some of the cultural adaptations that were needed?

o Could you also provide a little more information on how the original training in India was developed?

- Results and discussion

o Were there any individuals that started but did not finish the course?

o Did participants share any perspectives on the time commitment to complete the course?

o Around half of participants did have mental health training and many of them also had advanced degrees, which is not likely representative of non-specialists who may be taking on psychological care. Those demographics would be appropriate to more clearly report in the results, instead of just referring readers to the table, and comment on as a limitation.

o Were there any notable differences in how participants responded to the foundational course and the BA course?

---

## [Reviewer Report]

Thank you for inviting to review this well-written manuscript for development of a fully digital program to address treatment gap in the rural America. I look forward to reading about further work on this domain. Please see few minor points for consideration:

1. Title: Is it development of a digital training program or a digital platform? Digital platform means software development and the text itself refers to digital content creation than a digital platform, so it seems more like development of a digital training program.

2. In introduction, lines 37-40 can be moved up to explain the reasons for low proportions of persons receiving psychotherapy and build rationale for expanding access (lines 17-21).

3. On page 5 of a link for the Healthy Activity Program Manual can be given would be good (line 28).

4. On page 6, in blueprinting step, it would be helpful to know the qualifications and experience of the team members who created the course blueprint as well as the process through which certain core competencies were derived from the learning objectives. Additionally, were the reviewers specialized in the Healthy Activity Program and what process was used to finalize the blueprint?

5. On page 8, in video production, it is mentioned that production time was 4 months but it adds up to 6 months including 2 months for pre-production, 1 month for filing and 3 months for video/sound editing (line 30). On page 11 as well, it says total of 6 months for course development (lines 43-44). It is not clear if there are 2 separate things being talked about here.

6. On page 9, it would be helpful to know what kind of Leaning Management System has been used for the purpose of current study.

7. On page 10, just curious to know the reason for collecting the socio-demographic information after course completion.

8. On page 11-12, could it be clrafified, what is the difference between module and lessons as there seem to be more lessons than modules.

9. On page 12, can it be specified, how was the time taken for course determined (through FGD, or any other means). Could trainees move between different modules, did the assessments have a bearing on course completion (how much they scored, did they pass/fail)

10. On page 13, how much lag was there between the 2 courses (lines 1-2), what does being able to scroll through the content mean and why was it only in BA (lines 27-28). Could it be that only those who were more committed stayed and had also gotten more familiar with the course that they liked BA more?

11. Participant feedback mentions adequate time given, can more details be added on the same in the manuscript. Before beginning, were participants given or suggested a timeframe in which to complete the course, how was it monitored or enforced?

12. Participant feedback: general layout says there were live enactments in Behavioral Activation and that “it was nice to see real people presenting the content”. I did not get a difference between two modules in the main text, seemed both had actors playing counsellors in lecture and role play videos as well.

13. While free response questions have been added, what will be the methodology for checking their accuracy?

---

## [Reviewer Report]

I really enjoyed reviewing this manuscript - thanks to the authors for a compelling piece. The authors describe the process that they underwent to develop the tools clearly, and provide a strong and coherent rationale for the core intervention components that they selected for the training. Overall, I feel that this study makes important contributions to bridging the gap in community mental health care access through task sharing models and non-specialist provided psychosocial care.

I would be interested to learn more about the UI/UX of the LMS and maybe to see any design artifacts (e.g., diagramming user needs or or other design thinking activities they engaged in), but this is likely outside of the scope of the present study.

---

## [Reviewer Report]

This paper summarizes the developing and adapting of a digital training program for training non-specialists in Texas, USA.

The topic is important, and the innovation is useful. The paper could be strengthened as follows:

1. would be useful to add a citation for BA when it’s first introduced in the intro

2. Authors could simplify the intro by bringing some of the Texas background, including a specific depression prevalence rate, into the intro. Similarly, there is some repetition from the intro, the setting, and the intervention description - esp when introducing nonspecialists. authors could introduce their nonspecialist concept and target population early on and continue with this rather than reintroducing it in each section

3. please simplify throughout the paper. there are too many complicated/wordy sentences, and some run-on sentences.

4. when digital training in US is mentioned itd be useful to give a brief example of what this model looks like currently for CHWs eg do they use only online modes, combo with in person? especially as currently it seems only used for CE (post training) rather than initial training

5. methods - the paper describes adapting this to rural context in Texas, but there is no description of the cultural adaptation process for the manual to properly represent Texans. could the authors speak to this?

6.results section could be improved with subheadings. Also, I think moving the methods for the formative testing from the results (line 37-46) into the methods section would offer better flow.

7. can the authors speak to the relevance of course content—the cadre of participants had some background in MH, and CHW programs in US typically involve some form of basic patient care in their curriculums. Did participants find all content relevant, esp the Foundational courses?

7. the discussion currently has at least two paragraphs worth of summarizing the results. rather than summarizing results, the authors could elaborate the discussion more in terms of relating to the broader audience and initiatives in US as well as globally. Similarly, I think a recommendation section or future steps might help to organize this and promote cohesion in any other initiatives that may be ongoing. here are some questions/thoughts I had, one or some of which could be addressed in the discussion:

-When will the CHWs have time to participate for 8-10 hours a week for 6 weeks? Recommendations needed for how this could be potentially integrated into current systems and what needs to be investigated to determine any shortfalls or barriers

-what about safety during this online course – is there an immediate contact for users if a learner is distressed by any of the content? Eg, topics dealing with suicide assessment, given the increase of those impacted during COVID.

Also, any immediate contact for issues with the digital program? what will this look like if scaled (costs etc), are there existing formats in the US this study could pull from?

-participants mentioned this would be an initial start to training, but on what exactly, the depression care? And how might they imagine the extended training they need in the future, e.g., when, how long, how often; digitally, in person or both?

- would these results look different for CHWs with no mental health background? Will authors investigate this before implementation, or will this platform be only geared to those with some mh background? If so, how could this background be determined/assessed?

-similarly, what kind of nonspecialists or CHWs might this be useful for in Texas or more broadly in US (eg working in obstetrics, general health)-- any recommendations from these findings or for future research?

-The authors mention the significance of achieving competency. There is a competency framework/platform that is rapidly growing in the global mental health field, the WHO/UNICEF EQUIP platform. I am curious whether the authors find this platform helpful in their future search for establishing competency, and/or whether it’s useful to align with global initiatives like this one, or others?

---

## [Reviewer Report]

The authors describe their efforts to develop a digital training platform for non-specialist providers in the United States. The paper is well-written, and the objective of the project is an important one. The authors note this study involved replicating prior efforts from rural India. As such, much of my comments are centered around ensuring this manuscript represents a unique contribution to the literature.

Abstract:

- The final clause of the opening sentence reads : “particularly in the US.” However, neither in the abstract nor the main body do the authors expand on the particularity of the US in challenges to scaling up task sharing. I would suggest removing this clause or retaining this after edits to the introduction (should the into be edited in line with suggestions below).

Introduction

- Throughout the introduction, I had some questions regarding citations, and if the provided citations were the most appropriate for points being made. For instance, the authors state: “This is a major concern because in addition to strong evidence supporting the clinical effectiveness of psychosocial interventions (Kazdin 2017)…” However, the Kazdin article is a perspective piece on challenges with the dominant mode of care delivery. A review or meta-analysis on the effectiveness of interventions may be more appropriate here.

o Spot checking the next references in that sentence, “patients overwhelmingly express preference for psychosocial interventions over pharmacological treatment (Dwight-Johnson et al. 2001; McHugh et al. 2013), the McHugh meta-analytic review seems appropriate but the Dwight-Johnson article is an older examination of the impact of QI programs on delivering care consistent with patient preferences. This seems distinct from the claim in the paper, particularly when the authors have another appropriate citation that more directly addresses the points made.

- The authors point that the inequal distribution of providers across the US impacts the care of racial and ethnic minority groups is an important one. First, I would caution the authors again about citations and suggest that citations be moved to match their claims. The Dinwiddie and Le Cook articles do reference racial and ethnic inequities; however the Andrilla and Larson citations are about geographic distribution and should be separated and cited after that clause.

o Additionally, with regard to this point, while I agree it is an important consideration, it’s incorporation in this paper seems like an after-thought. Given the potential of task sharing to address racial and ethnic inequities, more attention to this point throughout the introduction and/or discussion would be a nice addition.

- The authors describe the emergence of digital technologies for training non-specialist providers. They also describe their work as a replication of prior efforts in rural India. As such, as a reader I was left to wonder “so what is the novel contribution of this paper?” I do think that this paper can make a novel contribution; however, I think the authors should spend some time in the introduction describing exactly the gap in the literature that this paper address and the uniqueness of this project, particularly given that it does replicate prior efforts. Note, I think replications of prior studies are very important and applaud the authors on replicating their work—but thinking about the value added by this paper, I think more commentary on the unique considerations of this replication are important.

- I was confused by the following citation – “We describe the development and initial formative testing of a digital curriculum (Patel et al. 2022)…” I can assume the connection to the EMPOWER work, but it is not immediately clear why that paper should be cited there.

Methods:

- In line with highlighting the unique contribution of this manuscript, further description on how the Healthy Activation Program was adapted to fit this context would be interesting. It is novel that interventions used in LMIC are being transported to the US – describe the process of that transportation in greater depth.

- A key consideration of human-centered design is involving users throughout the development process. A limitation of the reported approach is that users were not involved until the testing phase. Additionally, the targeted users do not exactly overlap with the eventual end-users of the intervention, due to the prioritization of health workers and those with experience delivering mental health care. This should be noted as a limitation, as these groups likely differ in their needs and preferences.

- A larger comment on methods, the authors note user testing was to ensure/understand acceptability and feasibility. I would suggest the authors investigate the concept of usability and consider if that concept more accurately describes their testing. I do not think the methods investigate acceptability (general sense of satisfaction) or feasibility (practicability or suitability for everyday use).

- Standards for usability testing and development may also be helpful when thinking through what other data might also need to be reported and what the limitations of this project may be.

- Including the focus group guide as a supplemental material would be helpful.

- Was transcript coding done to consensus? Was coding completed separately? A more thorough description of the qualitative procedures would be helpful.

Results

- The total time necessary to complete both courses is was quite long. That did make me wonder about the overall feasibility of having learners complete the program. It also made me wonder how much of the program your user testing participants might have completed. Are there any engagement statistics from the user testing participants? How did they access the program (phone or computer)? Were participants compensated for their participation in the study?

- It would be helpful to separate the 12 participants who completed the Behavioral Activation course in the demographics Table 2. Did these participants differ from the 21 who completed the Foundation Skills course in any meaningful ways?

- Please include a breakdown of how many participants participated in each focus group. Please also include demographic/identity information on who conducted the focus group. I’d suggest consulting a qualitative reporting checklist to ensure relevant information is reported throughout.

- Were there any participant suggestions that were not actionable? Were there any higher level comments on the acceptability and feasibility of engaging in a longer form training completely online?

Discussion:

- The authors note “the need to engage the target audience throughout the development process;” however, as noted above, their engaged was limited to after a curriculum had been developed and uploaded into an LMS. This may be interesting to discuss in greater depth, as often times it is hard to balance user input at all stages with the needs/expectations of research projects.

- In parts the discussion reads like a recap of results, such as the paragraph on participant feedback on page 16. The authors should identify opportunities to further discuss the nuances of their process, the unique considerations of doing this work, what sets this project apart from other training development studies (including their own prior work).

- In the first paragraph on page 17, the authors note “The community health workers emphasized barriers to addressing mental health concerns in their communities, including limited training opportunities, and emphasized the need for needs-based training programs to build their skills in mental health care (Garcini et al. 2022). Therefore, the training program developed in this study could bridge this gap, and support capacity building of non-specialists including community health workers.” However, the relevance of this paper to this point is not immediately clear. How were the unique needs of community health workers addressed in your methods? How was this potential applicability considered in this study? Of note, there are also other papers documenting “interest in deploying community health workers for addressing mental health challenges in historically underserved communities.” Much of this work also occurred prior to COVID-19, including a review paper https://link.springer.com/article/10.1007/s10488-017-0815-0

- With regard to limitations, I would suggest the authors focus more on the limitations of the described approach (e.g., user engagement in later phases, limited information on the statistics of user engagement leading to questions if participants reflections are on the entirety of the course, not doing user engagement and testing on individual modules, etc). I think more attention paid to the limitations of this study and its methods, instead of the conclusions to be dran from this study, would help to better conceptualize results.

---

## [Editor Report]

Thank you for submitting your manuscript for review. Although the reviewers acknowledge the relevance of the subject, they have identified notable flaws in the methodology, findings, and their interpretation. The reviewers have provided useful recommendations that could improve the manuscript. We invite you to carefully consider and address the reviewers’ comments and recommendations and submit a revised manuscript.

---

## [Reviewer Report]

I congratulate the authors for their innovations in scaling up the delivery of psychosocial interventions. They have successfully addressed all queries and made suitable edots to be manuscript. I do not have any further comments and look forward to reading more of their work in this direction

---

## [Editor Report]

Thank you for revising the manuscript and responding to the reviewers’ recommendations. The reviewers are satisfied with the revisions and have recommended that we accept the manuscript. We are happy to accept the manuscript in its present form and look forward to working with you through the publication process.